# Targeting Cardiac Stem Cell Senescence to Treat Cardiac Aging and Disease

**DOI:** 10.3390/cells9061558

**Published:** 2020-06-26

**Authors:** Eleonora Cianflone, Michele Torella, Flavia Biamonte, Antonella De Angelis, Konrad Urbanek, Francesco S. Costanzo, Marcello Rota, Georgina M. Ellison-Hughes, Daniele Torella

**Affiliations:** 1Department of Medical and Surgical Sciences, Magna Graecia University, 88100 Catanzaro, Italy; cianflone@unicz.it; 2Department of Translational Medical Sciences, AORN dei Colli/Monaldi Hospital, University of Campania “L. Vanvitelli”, Via Leonardo Bianchi, 80131 Naples, Italy; michele.torella@unicampania.it; 3Department of Experimental and Clinical Medicine and Interdepartmental Centre of Services (CIS), Magna Graecia University, 88100 Catanzaro, Italy; flavia.biamonte@unicz.it (F.B.); fsc@unicz.it (F.S.C.); 4Department of Experimental Medicine, Section of Pharmacology, University of Campania “L.Vanvitelli”, 80121 Naples, Italy; antonella.deangelis@unicampania.it; 5Molecular and Cellular Cardiology, Department of Experimental and Clinical Medicine, Magna Graecia University, 88100 Catanzaro, Italy; 6Department of Physiology, New York Medical College, Valhalla, NY 10595, USA; marcello_rota@nymc.edu; 7Centre for Human and Applied Physiological Sciences and Centre for Stem Cells and Regenerative Medicine, School of Basic and Medical Biosciences, Faculty of Life Sciences & Medicine, King’s College London, Guys Campus-Great Maze Pond rd, London SE1 1UL, UK; georgina.ellison@kcl.ac.uk

**Keywords:** senescence, aging, adult stem cells, tissue homeostasis, epigenetics, metabolism, SASP

## Abstract

Adult stem/progenitor are a small population of cells that reside in tissue-specific niches and possess the potential to differentiate in all cell types of the organ in which they operate. Adult stem cells are implicated with the homeostasis, regeneration, and aging of all tissues. Tissue-specific adult stem cell senescence has emerged as an attractive theory for the decline in mammalian tissue and organ function during aging. Cardiac aging, in particular, manifests as functional tissue degeneration that leads to heart failure. Adult cardiac stem/progenitor cell (CSC) senescence has been accordingly associated with physiological and pathological processes encompassing both non-age and age-related decline in cardiac tissue repair and organ dysfunction and disease. Senescence is a highly active and dynamic cell process with a first classical hallmark represented by its replicative limit, which is the establishment of a stable growth arrest over time that is mainly secondary to DNA damage and reactive oxygen species (ROS) accumulation elicited by different intrinsic stimuli (like metabolism), as well as external stimuli and age. Replicative senescence is mainly executed by telomere shortening, the activation of the p53/p16^INK4^/Rb molecular pathways, and chromatin remodeling. In addition, senescent cells produce and secrete a complex mixture of molecules, commonly known as the senescence-associated secretory phenotype (SASP), that regulate most of their non-cell-autonomous effects. In this review, we discuss the molecular and cellular mechanisms regulating different characteristics of the senescence phenotype and their consequences for adult CSCs in particular. Because senescent cells contribute to the outcome of a variety of cardiac diseases, including age-related and unrelated cardiac diseases like diabetic cardiomyopathy and anthracycline cardiotoxicity, therapies that target senescent cell clearance are actively being explored. Moreover, the further understanding of the reversibility of the senescence phenotype will help to develop novel rational therapeutic strategies.

## 1. Introduction

Several stressful insults and certain physiological processes trigger cellular senescence, which is mainly characterized by a stable and largely irreversible replicative blockage accompanied by secretory features, macromolecular damage, and deregulated metabolism [1] (Figure 1). In other words, cellular senescence is a intricated phenomenon that profoundly changes the phenotype and function of proliferative cells [2]. More than five decades ago, Leonard Hayflick demonstrated that primary human cells sub-cultivated in vitro have a limited proliferative capacity [3], such that cell cultures stop dividing after an average of 50 population doublings; this proliferation defect over time, named replicative senescence, represents the first hallmark of cellular senescence. Replicative senescence is clearly related to time and is therefore a major feature of aging [4]; it is classically associated with telomere shortening, the progressive uncapping of chromosomes at each cell division [5]. Replicative senescence is linked to a multi-component senescence-associated secretory phenotype (SASP), which is considered the second hallmark of senescence [6,7]. Assuming that human cells have on average a lifespan of 50 population doublings, 2^50^ is more than enough cells for several lifetimes. However, as aging advances and several age-dependent and age-independent pathologies develop, cell senescence is caused and/or fostered by several stress and physiological stimuli, and it can have context-dependent positive or detrimental effects on the organism [8]. Indeed, several senescence-inducing stressors cause genomic or epigenomic changes, an oncogenic transformation threat for cells and tissues [9]. Thus, senescence arrest is classically viewed as an endogenous protection from cancer development. Concurrently, the SASP fine-tunes relevant physiological processes such as tissue repair and the formation of specific embryonic structures [9,10]. On the other hand, senescent cells can dictate the loss of tissue homeostasis, thus reducing the regenerative and reparative capacity of a tissue mainly due to the functional decline of its stem cell compartment and dysregulating the normal function of adjacent cells through SASP-dependent cell-to-cell communications.

Aging classically alters many tissues of vertebrate organisms through the progressive accumulation of senescent cells [11,12]. Replicative senescence that results in cell cycle arrest has been generally envisioned as a highly static state that, as said above, is contrary to the highly active process related to the SASP [2]. This apparent contradiction seems to be overcome when considering that cell cycle arrest is not such a permanent and static state for senescent cells. Senescent cells, indeed, possess an active metabolism, that differentiates them from apoptotic and quiescent cells [2,8]. Accordingly, a typical feature of senescent cells is a progressive enlargement despite a lack of cell division. The SASP accounts for the high metabolism of senescent cells that produce and secrete potent biological factors that mainly act on the neighboring cells and surrounding tissue but also at distance when offloaded in the systemic circulation. The SASP activities promote inflammation, invasion, angiogenesis, and, counterintuitively, cell proliferation. The SASP’s properties have been used to envision how a relatively small number of senescent cells can have such profound local and systemic effects in fostering aging and age-related pathologies. Hence, cell-to-cell communications—mainly through the paracrine effects of secreted factors by the SASP affecting surrounding cell and tissue—better explains the participation of cellular senescence in various physiological and pathological processes in contrast to the lone proliferative arrest [13]. In essence, senescent cells may profoundly affect tissue homeostasis, interfere with organ function, instruct other cells in their environment, or evoke secondary amplified immune network responses, which are highly dynamic processes with a potential selective advantage for tumor growth.

Adult stem/progenitor cells are a small population of cells that reside in tissue-specific niches and possess the potential to differentiate in all the cell types of the organ in which they operate. Adult stem cells (e.g., hematopoietic stem cells, intestinal stem cells, satellite cells, skin stem cells, and germline stem cells) are implicated with the homeostasis, regeneration, and aging of practically all tissues [14,15,16,17,18,19]. Adult stem cell regenerative function declines with age, and cell-autonomous and non-autonomous modifications within in the niche, as well as in the circulating blood, are involved in this age-related decline [20,21,22]. In organs characterized by high and moderate turnover (like gut, bone marrow and skin), aging appears to be dictated by adult stem cell senescence [20,21,22]. In the reverse, in organs with a low cell turnover, like the brain and the heart, aging is mostly related to parenchymal cell senescence [23,24]. However, brain and heart stem cell senescence contribute to the respective organ failure with aging [25,26]. Aging is indeed a classical irreversible risk factor for the development of cardiovascular diseases, the prevalence of which considerably increases as age advances [24]. Age is associated with cellular and molecular changes in cardiac tissue homeostasis and response to injury, thus resulting in the progressive deterioration of the structure and function of the heart [24]. Accordingly, tissue-resident cardiac stem/progenitor cells (CSCs) undergo cellular senescence through augmented reactive oxygen species (ROS) formation, oxidative stress, and the loss of telomere/telomerase integrity in response to several physiological and pathological stimuli with aging [27,28]. Aged-senescent CSCs contribute to impaired heart regeneration, which is secondary to the SASP of CSCs that dictate otherwise healthy CSCs to undergo senescence [29]. The elimination of senescent CSCs using senolytic drugs abolishes the SASP and its detrimental consequences in vitro and in vivo, thus rejuvenating the aging dysfunctional heart [29].

On this premise, in this review, we first describe the types of the intrinsic factors that damage adult stem cells during aging, as well as the extrinsic factors, such as the SASP, that contribute to the functional deficit of adult stem cells during aging. We then mainly discuss what it is known about CSC senescence during aging and aging-associated diseases. We propose that intrinsic and extrinsic CSC senescence contributes to age-related decline in the maintenance of cardiac anatomy and function, as well as the development of cardiovascular diseases, including diabetic cardiomyopathy and chemotherapy-induced cardiomyopathy. Though this review article indirectly deals with cardiac senescence and aging, the focus is on CSCs, so readers are thus referred to recent review articles for further elaboration on the aging of the heart and of other cardiac cell lineages [24,30]

## 2. Molecular Mechanisms of Adult Stem Cell Senescence and Aging

The main intrinsic and cell-autonomous molecular mechanism of the senescence program regulates cell cycle arrest (Figure 2). In mammalian cells, the retinoblastoma (RB) family of pocket proteins and the tumor suppressor p53 play a key role in the senescent cell-cycle inhibition [31]. RB1 and its relatives p107 (retinoblastoma-like 1—RBL1) and p130 (retinoblastoma-like 2—RBL2) are phosphor-proteins targeted by specific cyclin-dependent kinases (CDKs). RB protein phosphorylation de-represses E2F family transcription factor activity, which is key for cell-cycle progression [32]. Additional key regulatory proteins including the p21 (Cdkn1a), p16^INK4A^, and p19^ARF^ (both encoded by the Cdkn2a locus) cell cycle inhibitors are increased in senescent cells, thus contributing to the inhibition of the cell cycle in G1-S fostering an irreversible arrested state [33,34,35,36]. Non-coding RNA-mediated and microRNA, in particular, gene silencing also regulate replicative senescence [37].

Telomere shortening is a hallmark of stem cell aging [1,5,21]. Indeed, the telomeres of several tissue-specific adult stem cells shorten with age, despite possessing telomerase [38,39]. Telomerase reverse transcriptase (TERT) overexpression in mice increases their median lifespan [40,41]; however, it is unknown whether the latter depends on the reversal of adult stem cell senescence and aging. The standard laboratory mice begin life with much longer telomeres than human beings, but they have shorter lifespans. Mice lacking the telomerase RNA component (TERC) are, phenotypically, almost indistinguishable from their wild type counterpart for five consecutive generations. However, stem cell attrition becomes apparent at the sixth generation when these mice reach critical telomere length, although at the fourth generation, it already becomes evident a skewed hematopoietic stem cell (HSC) lineage potential phenotype [42,43].

The DNA damage response (DDR) to either intrinsic or external insults often initiates replicative senescence and the progressive accumulation of DNA damage; the ensuing mutations characterize adult stem cell senescence [44]. Histone H2AX phosphorylation and comet tails, typical DNA damage markers, increase with age in HSCs [45,46], as well as in satellite cells [47]. Moreover, HSCs from old subjects show signs of replication stress and the downregulation of DNA helicases, making them more vulnerable to replication problems [48]. The oncogenic transformation of adult stem cells is a risk that increases at each stem cell DNA replication and division. Accordingly, the lifetime risk for any tissue to develop a cancer is proportional to the number of divisions its resident stem cell compartment has undergone [49]. Quiescence is a state of reversible growth arrest that appears to be a self-protective state for several tissue-specific adult stem cells. Adult stem cell prolonged quiescence has been classically interpreted as a dormant, low-activity state, but it has also been envisioned as a state of active ‘self-control,’ as stem cells “*idle*” steady before moving into activation to proliferate and differentiate when needed [50]. Quiescent cells are inherently safeguarded from replicative damage in consideration of their dormant/low-activity state. However, if DNA damage occurs, double-strand breaks (DSBs) in quiescent cells, i.e., in the G0–G1 phase of the cell cycle, are more likely to be repaired by mutation-prone non-homologous end joining (NHEJ) rather than by a more accurate repair mechanism such as homologous recombination (HR) [51], thus generating mutations. Indeed, quiescent HSCs depend more on NHEJ to repair DSBs, whereas proliferating HSCs depend more upon HR [44,52]. Nevertheless, when adult stem cells are actively proliferating, they are more prone to accumulate DNA damage [53], despite proliferating stem cells being able to repair DSBs more efficiently than when they reside in quiescence. As time goes by, normal tissues get more somatic mutations [54]. Most these mutations do not alter cell and tissue function, so the large majority of somatic mutations are not capable of selection. However, a mutation conferring a selective growth advantage to the mutated stem cell can rarely occur. The mutation-selected stem cell and its progeny behave then as a “clone,” progressively expanding over time and, in particular, in the elderly [55]. The latter has been clearly shown to happen in the HSC compartment, resulting in a process called “clonal hematopoiesis” where the mutated stem cell clone gives rise to a considerable amount of mature blood cells. These DNA mutations confer a proliferation-/survival-advantage and promote HSCs with precancerous clonal expansion [54,55,56,57,58]. On the other hand, DNA damage during aging may also select mutations that increase senescence, apoptosis, and/or differentiation, resulting in a reduced number of tissue-specific adult stem cells.

Adult stem cell function is appreciably regulated by chromatin state, which is another important mechanism for cell senescence. The epigenetic landscape envisioned by C. Waddington [59], in essence, depicts the dynamic flow of gene communications that restrict and constrain cell fate down the developmental slope [60], thus representing a metaphor for the qualitative understanding of the developmental processes of cells and tissues. Gene loci that are key for cell fate decisions are indeed bivalent; in other words, they concurrently harbor both active and repressive chromatin modifications [61]. Aging accounts for several chromatin and gene expression changes. DNA methylation, which is a repressive epigenetic mark, decreases with age in HSCs, although this repressive change is variable. This hypomethylation is closely connected to the proliferative history of HSCs, offering an indirect explanation of their persistence in the quiescence state for prolonged times. Tri-methylation at the fourth lysine residue of the histone H3 protein (H3K4me3), an activating modification, increases with age at gene loci that control HSC self-renewal, theoretically explaining the increased number of HSCs often encountered in aging [62]. Furthermore, the amount of the acetylation at the 16th lysine residue of the histone H4 protein (H4K16Ac), another activating change, decreases with age in HSCs [63]. On the contrary, in satellite cells, the levels of H3K4me3 are mildly reduced with age, while the amount of the tri-methylation at the 27th lysine residue of the histone H3 protein (H3K27me3), a repressive mark, considerably rises with age, which is concurrent with a reduction of the expression levels of the histones themselves with age [64]. Additionally, several chromatin modifiers—including components of the SWI-SNF and polycomb repressive complexes (PRCs), the Histone deacetylases (HDACs) including sirtuins, and DNA methyltransferases—change their expression levels with age in stem cells [65,66,67]; the latter suggests that alterations of epigenetic modifications are a general feature of stem cell aging that underlies their functional decline. Chromatin progressively acquires permanent changes as a function of progressive age of organisms and secondary to cell stress, in particular secondary to DNA damage signals. DSB has the greatest lasting effect on chromatin among all the distinctive forms of DNA damage. From yeast to mammals, chromatin factors are dramatically rearranged by DSBs as part of the reaction to damage that is not entirely repaired [68,69]. Accordingly, chromatin modifications in response to DNA damage seem to also mediate the skewed differentiation potential of aged adult stem cells [70].

In stem cells, the polycomb group (PcG) proteins, a particular family of transcriptional repressors, are key regulators of cell self-renewal and cell fate commitment [71,72,73,74,75]. PcG possess several stable gene repressing functions, among which these proteins establish a repressive chromatin structure, inhibit the chromatin remodeling system, prevent the transcriptional initiation organization and also inhibit the interactions of enhancer/promoter that enable transcription [76]. PRC1 and PRC2, two distinct PRCs, cooperate at the initial site of repression and induce epigenetic modifications of chromatin to enable gene silencing [77,78]. PRC1 includes Cbx, Mph, Ring, Bmi-1, and Me118. PCR2 includes Ezh2, Suz12, and Eed, and it initiates silencing by increasing histone H3 Lisine-27 (H3K27) methylation [79,80]. The methylation of H3K27 appears to drive PRC1 binding to chromatin, which mediates steady gene silencing [81,82]. Independently from their role in development as regulators of repressive chromatin states, PcG proteins also regulate stem cell self-renewal and growth. Bmi-1, a PRC1 member, is indeed essential for adult HSC self-renewal and amplification [83,84]. Bmi-1 targets the repression the INK4a-ARF tumor suppressor gene locus, encoding p16^INK4a^ and p19^ARF^ (called p14^ARF^ in humans), which are key players of the molecular mechanisms of senescence in human and rodent primary cells [85] (Figure 2). The over-expression of Bmi-1 in mouse embryonic fibroblasts significantly decreases the expression levels of p16^INK4a^ and p19^ARF^, and it bypasses replicative senescence promoting cell immortalization [86,87]. Human fibroblasts undergoing replicative senescence are characterized by the downregulation of Bmi-1 protein levels with the concomitant accumulation of p16^INK4a^ and p19^ARF^ proteins [88]. These data generated the current hypothesis that the age-related functional decline of tissue-specific stem cells is mediated, at least in part, by an increase in both cellular surveillance and tumor suppressor activity [89].

In another account, ROS, prototypical senescence inducers, were found to contribute to the decline in stem cell function and tissue repair during aging [90,91,92,93,94]. As age goes by, cell damage accumulates while mitochondrial integrity declines, thus leading to increased ROS formation and resulting in a vicious cycle that further harms cellular macromolecules and alters mitochondrial oxidative phosphorylation, eventually leading to cellular disarrangement [95]. Human mesenchymal stem cells (MSCs) increase ROS levels during aging [96], and the number of healthy hematopoietic stem cells (HSCs) with low ROS levels decreases with age in mice [97]. A mouse hematopoietic system genetically modified to lack the transcription factors FoxO1, FoxO3, and FoxO4, which are downstream effectors of the insulin and insulin-like growth factor 1 (IGF-1) signaling pathways, harbors HSCs with accumulated ROS, resulting in their exit from quiescence followed by apoptosis and a severe deficit in their repopulating function [98]. Additionally, the erasing of phosphatase and tensin homolog (PTEN) and the concomitant ablation of protein kinase B (AKT) 1 and 2 lead to a substantial deficit in long-term HSC repopulating potential. This evidence envisions a molecular scenario where the PTEN–AKT–mammalian target of rapamycin (mTOR) pathway, upstream of FoxO, modulates ROS levels due to the regulation of HSC self-renewal and survival [99,100,101]. Furthermore, as age advances, FoxO3 and the DNA damage sensing serine/threonine protein kinase ATM (ataxia-telangiectasia mutated) modify superoxide dismutase 2 (SOD2), thus contributing to the functional decline of aged HSC [102,103,104]. Accordingly, the sirtuin family of NAD-dependent protein deacetylases contemporarily regulate aging, oxidative stress, adult stem cell function, and sirtuin 1 (SIRT1), in particular, mediates the age-dependent deficit in MSC proliferation and commitment [105,106]. Additionally, HSC aging is regulated through ROS production by SIRT3 [107,108].

Finally, there is increasing evidence that cell metabolism is strongly associated with aging and adult stem cell senescence [109]. Indeed, decreased nutrient signaling can extend the lifespan, aging is accelerated by anabolic signaling, and the organismal lifespan is extended by several pharmacological manipulations of specific metabolic pathways [110,111]. The epigenetic states are altered by cellular metabolic pathways, and organismal aging and longevity are accordingly affected by these changes [112,113]. A metabolic clock involving mitochondria and nutrient-sensing pathways is involved in the aging process. With age, somatic mutations progressively accumulate in the mitochondrial DNA (mtDNA), and because mitochondria are exposed to an oxidative environment, mtDNA lacks both protective histones binding, and an efficient repair mechanism. Concurrently, mtDNA mutations accumulate with high frequencies in adult stem cells with aging [114]. Nutrient sensing dysregulation is one of the major hallmarks of aging and nutrient-sensing systems along with the ‘‘insulin and IGF-1 signaling’’ (IIS) pathway, mTOR, AMP-activated protein kinase (AMPK), and sirtuins, all of which have been linked to aging [108]. Mitochondrial metabolism and nutrient-sensing pathways are regulated by caloric restriction. Interestingly, caloric restriction increases the lifespan and/or healthspan in many eukaryote species, including non-human primates [110,115,116]. Overall, modifications in metabolism secondary to environmental stimuli appear to globally affect the epigenome of adult stem cells, thus inducing their senescence. Different metabolites change or keep specific epigenetic states, thereby evoking permanent alterations in gene expression, regulating stem cell fate and senescence in particular [109].

Growing evidence suggests that cytosolic-free iron triggers the molecular network underlying the development of cellular senescence. Intracellular iron content exponentially increases during cellular senescence, reaching approximately 30-fold higher levels than young cells and therefore contributing to increased oxidative stress and cellular dysfunction [117]. Accordingly, the expression of iron homeostasis proteins is altered in senescent cells. Redox-active iron accumulates in excess in the cytosol as a consequence of defective ferritin, promoting: 1) the increase in the translation of iron-dependent ferritin; 2) a reduction in ferritinophagy secondary to a decrease in the nuclear receptor coactivator 4 (NCOA4) due to DNA instability; and 3) an upregulation of ROS, which promotes oxidative injury that results in DNA damage and the oxidation of both lipids and proteins. The latter results in cells either activating the molecular network (which could be seen as a self-defense response) or, as an alternative, increasing their death by ferroptosis. A recent study demonstrated that human fibroblasts and neurons exposed to non-ferritin-dependent iron undergo both cell senescence and cell death by ferroptosis, suggesting that iron is a key player in the mechanisms leading to aging and neurodegeneration [118]. CRISPR/Cas9 technology was employed to obtain induced pluripotent stem cells (iPSC) from neuroferritinophaty (NF) patients. Fibroblasts, neural progenitors, and neurons derived from these NF-iPSCs possess high levels of cytosolic-free iron, which is associated with alterations in iron parameters, ferritin aggregates, oxidative damage, and the development of a senescence phenotype [118]. Altogether, these data postulate a role for iron metabolism and ferritin in stem cell senescence and aging.

## 3. SASP and Stem Cell Senescence and Aging

Senescent cells produce and release a variety of factors, including pro-inflammatory cytokines (IL-6; IL-7; IL-1; IL-1b; IL-13; and IL-15) and chemokines (IL-8; GRO-a, -b, and -g; MCP-2; MCP-4; MIP-1a; MIP-3a; HCC-4; eotaxin; eotaxin-3; TECK; ENA-78; I-309; and I-TAC), growth and angiogenic factors (amphiregulin; angiogenin; EGF; epiregulin; bFGF; heregulin; HGF; IGFBP-2, -3, -4, -6, and -7; KGF; NGF; PIGF; stem cell factor (SCF); SDF-1; and VEGF), matrix metalloproteinases (MMP-1, -3, -10, -12, -13, and -14; TIMP-1; TIMP-2; PAI-1, -2; tPA; uPA; and cathepsin B), receptors/ligands (EGF-R; Fas; ICAM-1, -3; OPG; uPAR; SGP130; sTNFRI; sTNFRII; and TRAIL-R3), non-protein molecules (nitric oxide; PGE2; and ROS), and insoluble factors (collagens, fibronectin, and laminin), all together unified under the name of the SASP [6,119]. The SASP constitutes the extrinsic arm of cell senescence mediating many of their non cell-autonomous patho-physiological effects [1] (Figure 2). Cell cycle arrest and the SASP are differently regulated, as, indeed, the activation of the p53 and p16^INK4A^/Rb tumor suppressor pathway underlies replicative senescence while the SASP is regulated by the activation of specific transcription factors such as the C/EBPβ, GATA4, NF-κB, mTOR, p38MAPK, and Notch1 signaling molecules [119,120,121,122,123,124,125]. Additionally, the SASP is controlled by a significant epigenetic regulation, whereby MLL1 (KMT2A), HMGB2, H2A.J and MacroH2A relocate soon after the induction of senescence to induce the transcriptions of SASP genes [126,127,128,129,130].

The main function of the SASP is to reinforce cell senescence by secreting pro-senescent cytokines such as IL1, IL6, and IL8 [130,131,132]. Furthermore, the SASP induces paracrine senescence in neighboring cells [133]. On the other hand, Csf1, Ccl2, and IL8 (possibly Cxcl-3 in mice), constituents of the SASP, are key attractant signals for immune cells, and macrophages and natural killer (NK) cells in particular, which act to eliminate senescent cells [134,135,136]. Recently, additional functions of the SASP have also been revealed, as the SASP secretome can trigger proliferation, angiogenesis, or epithelial–mesenchymal transition (EMT) in neighboring or cancer cells [6,137,138]. Remarkably, the SASP exerts reprogramming-like functions, as it has been found that transient exposure to the SASP modifies primary mouse keratinocytes to increase the expression of stem cell markers that show regenerative potential in vivo [139]. Additionally, the expression of stem cell markers in the liver is boosted by the induction of senescence in single cells in vivo. Overall, these data shows that the SASP could also mediate a positive process by triggering stem cell plasticity and tissue repair; the latter has generated the tantalizing hypothesis that the local injection of senescent cells could be a novel regenerative approach to foster tissue functional repair in vivo [139].

However, the SASP can contribute to stem cell dysfunction in aging and age-associated diseases [7]. Indeed, inflammaging and the increased secretion of SASP-related molecules can modulate stem cell dysfunction, which could be counteracted by anti-inflammatory SASP-inhibitors such as JAK/STAT antagonists [140]. Accordingly, genetic and pharmacologic approaches to specifically target and kill chronic and SASP-producing senescent cells have already been shown to increase the lifespan and to ameliorate the quality of life and disease severity [141,142,143,144,145,146].

Finally, there are burgeoning data that show that senescent cells produce and secrete extracellular vesicles (EVs), which differ in number and composition when compared to non-senescent cells, and these senescent cell EVs (SCEVs) play a key role in the detrimental effects of senescence in aging. SCEVs are loaded with microRNA, which has been recently shown to be part of the SASP and is specifically produced or retained by senescent cells. The analysis of the microRNA content of the SASP ha shown that there are selective microRNAs produced by the SASP, and these are mainly involved in the repression of pro-apoptotic genes [147]. These SCEV-microRNAs have been associated with various processes that govern the functional decline of tissue-specific adult stem cells in aging: cell senescence, stem cell number decrease, telomere erosion, and circadian rhythm [148]. SCEVs and their miRNA load are involved in the cell-to-cell communications via senescent cells, which can induce both a pro-fibrotic/inflammatory and pro-regenerative tissue response into neighboring cells [149,150]. Cell senescence specifies a precise repertoire of SCEV-derived microRNA within the SASP (e.g., the upregulation of miR-17-3p and miR-173 and the downregulation of miR-17-3p, miR-625-3p, miR-199b-5p, and miR-381-3p) that modulate an anti-apoptotic and, possibly, a pro-tumorigenic response in adjacent cell microenvironments [147]. Thus, depending on which type of SCEV-related microRNAs are released by the SASP, senescent cells have the ability to induce a tissue regenerative response from resident stem cells, though these SASP factors can also, on the contrary, impair stem cell regenerative actions in chronic conditions [139,141].

## 4. Cardiac Stem Cell Senescence

Since 2003, it has been reproducibly shown that the adult mammalian heart harbors a population of resident endogenous CSCs that participate in cardiac responses to injury and physiological CM turnover during the lifespan [151,152,153,154,155,156,157]. CSCs are clonogenic, self-renewing and multipotent, giving rise to a minimum of three different cardiogenic cell lineages (myocytes, smooth muscle, and endothelial cells), both in vitro and in vivo, that harbor significant cardiac tissue regenerative capacity [151,152,154,155,156,157,158]. Unfortunately, to date, there is still significant confusion and controversy over the endogenous role of CSCs as myogenic precursors contributing to myocardial homeostasis and repair/regeneration following injury [159,160,161,162,163]. The expression of c-kit (a type III receptor tyrosine kinase also named CD117 or SCF-R (stem cell factor receptor)) was instrumental in 2003 for the identification and characterization of endogenous CSCs in the adult mammalian heart [151]. However, in the adult myocardium, the detection of c-kit alone is inadequate and actually confusing when identifying true multipotent cells among all the other c-kit-positive (c-kit^pos^) cardiac cells. Indeed, the vast majority (~90%) of c-kit-labelled cardiac cells are endothelial and mast cells. Only less than 10% of the total c-kit-positive cardiac cells contain multipotent cells [154,155,157,158]. Sequential CD45/CD31 negative sorting followed by c-kit-positive sorting from total cardiac cells enriches multipotent CSCs, but this three marker-based prospective identification still identifies a heterogeneous cell population where only 10–20% of these CD45/CD31^neg^c-kit^pos^ cardiac cells are clonogenic/multipotent in vitro and in vivo [154,155,157,158]. Therefore, overall, only ~2% of the entire c-kit-positive cells fulfil the criteria for multipotent CSCs. This evidence suggests that c-kit as a sole target is indeed a poor biomarker for detecting CSCs within the adult myocardium. However, it is also fundamental to note that c-kit-negative cardiac cells do not harbor clonogenic/multipotent cells and c-kit haploinsufficiency reduces cardiomyocyte refreshment in the adult heart [155,158], which shows that c-kit identification alone is not sufficient but is still necessary to identify CSCs [154].

Overall, resident CSCs show a mixed and overlapping expression of several stem cell markers and an apparent multiplicity and heterogeneity of cardiac progenitor cell (CPC) sub-populations [164]. Thus, different CSC/CPC populations have been reported in the developing and adult heart: c-kit^pos^ CSCs [151,152,154,165,166]; cardiosphere-derived cells (CDCs) [153,167]; epicardium-derived cells (EPDCs) [168,169]; cardiac side population cells (SP) [170,171,172]; Sca-1^pos^ (stem cell antigen-1) CPCs [173,174,175]; Islet-1^pos^-expressing CPCs [176,177] and platelet-derived growth factor receptor-alpha (PDGFRα^pos^)-expressing CPCs [178]. Independently from the primary marker used for their original identification, these cardiac stem/progenitor cell populations are clonogenic, self-renewing, and multipotent, both in vitro and in vivo, and express specific transcription factors (Isl-1, Nkx2.5, MEF2C, and GATA4) in the embryonic and adult heart [164]. Moreover, these populations express several markers of stemness (Oct3/4, Bmi-1, and Nanog) and show significant regenerative potential in vivo [154,164].

Based on these data, a variety of studies have established that the heart contains a reservoir of stem and progenitor cells. Despite the “stemness” of a cell not being linked to a single specific biological marker, many reporting groups have independently described a “unique” CSC or CPC based on the primary marker used to isolate/detect them. With the exception of the Islet-1 cells, which dramatically decrease in number in adulthood [176] and seem to be remnants from the cardiac primordia [179], the identification of different cardiac stem and progenitor cells by the expression of precise membrane markers, which are, for the vast majority, overlapping among all the different described cell populations, suggest that these apparently phenotypically different cells are likely to be phenotypic variations of a unique cell type that is synthetically and overall described as CD31^neg^/CD45^neg^/c-kit^pos^/Sca-1^pos^/Abcg-2^pos^/PDGF-Rα^pos^ [154]. To avoid confusion, we keep the acronym CSCs to refer to all of the above cell populations that overall show similar regenerative/reparative/protective potential in vivo independently from the primary biomarker used to detect/characterize/isolate them [164].

The significant heterogeneity within c-kit labelled cardiac cells has prompted and spread confusion over the identity and regenerative role of endogenous CSCs. By targeting c-kit as the only marker, murine genetic fate map strategies based on the Cre/Lox recombination have been shown to be able to label more than 80% of c-kit-expressing cells in different known c-kit domains in the adult mouse [180,181,182,183]. Based on that premise, using these tools, the authors of the studies employing this technology have assessed the adult hearts by claiming that only a minimal number of cardiomyocytes derive from c-kit-expressing progenitors in adult life [162,180,181,182,183]. However, we have shown that this technology fails to fate map CSCs in the adult heart because only less than 10% of CSC-enriched CD45/CD31^neg^c-kit^pos^ are labelled in these c-kit^Cre^ mice [158]. Furthermore, CRE knock-in causes c-kit haploinsufficiency, which produces a significant deficit in the myogenic potential of CSCs in vitro and in vivo [155,158,161]. Therefore, appropriate and precise genetic fate map strategies that are able to actually label CSCs in vivo are still needed to address the myogenic role of CSCs in vivo.

The controversy and debate over the myogenic role of resident CSCs has been inadvertently fueled by the recent retractions of several papers by one of the scientists involved in the discovery and characterization of this cell entity [184,185]. It is a fact that the scandal surrounding those retracted publications has created a significant setback for the field of resident CSC biology and regenerative potential [184,185]. However, it must be cautioned that it would be equally devastating for this field if, because of those misdeeds of one investigator, all the independent and reproducible investigations of many other scientists on the regenerative role of CSCs were dismissed. It is worth outlining here that several independent groups have contributed to the characterization of adult resident CSCs [152], and these publications have never been questioned or retracted. Aside from the above scandal, which is not the topic of this review and is discussed elsewhere [184,185], it remains factual that clonal CSCs are robustly myogenic in vitro and in vivo [186]. The published record incontrovertibly shows that CSCs are potent myogenic precursors with significant cardiac remuscularization potential when transplanted in vivo [152,154,158]. Additionally, the cardioprotective and regenerative capacity of these cells lay on their paracrine activity through the direct secretion of growth factors or the production of extracellular vesicles [187,188].

Chronic heart failure is associated with a functional decline of the resident CSCs that progressively reduce their potential to preserve tissue homeostasis and to contribute new cardiac cells upon myocardial damage [189]. We and others have thus assessed whether the progressive accumulation of dysfunctional and senescent CSCs plays a crucial role in the pathogenesis of cardiac aging and failure [27,28,29,189,190,191,192]. Though cell senescence has been classically associated with the development of HF, even for this condition, an acute raise of senescent fibroblasts after myocardial infarction has been linked to a reduced fibrotic response of the myocardium, postulating a pro-regenerative effect from the acute and transient exposure to senescence program [96]. Nonetheless, burgeoning data consistently show that chronic exposure to senescent cells and the progressive accumulation of senescent stem cells inhibit the regenerative myocardial response to HF [27,28,29,189,190,191,192]. In line with the latter information, the selective clearance of senescent, p16^InkA^-positive cells that also accumulate in the mouse heart with aging reduces age-dependent cardiac hypertrophy and ameliorates the myocardial response to β-adrenergic damage [193].

Senescent CSCs, along with dysfunctional cardiomyocytes, accumulate with age in the myocardium, and this phenomena is directly linked to age independently from ischemic cardiomyopathy or other myocardial diseases like diabetic cardiomyopathy, hypertensive cardiomyopathy, valvular heart disease, and myocarditis [194]. Nevertheless, HF following chronic ischemic cardiomyopathy is associated with the senescence of left ventricular resident CSCs [195]. On this basis, the group of Antonio Beltrami isolated, propagated in culture, and analyzed CSCs from both recipients of cardiac transplantation affected by end stage HF and from normal donor hearts. They showed that both aging and HF decrease the pool of resident CSCs in human atria. Furthermore, age and pathology both induce CSC senescence and dysfunction, reducing their proliferative, migratory, and differentiation abilities [189,191,192,193,194].

We assessed in CSCs and cardiac myocytes from aging wild-type (WT) mice markers of cellular senescence, cell death, telomerase activity, telomere integrity, and regeneration [29]. To determine whether senescent program in the aging myocardium can be prevented, we compared the above data from WT mice with IGF-1 transgenic mice (TG), characterized by local myocardial IGF-1 overproduction [27]. p27^Kip1^, p53, p16^INK4a^, and p19^ARF^ expression in WT myocardial cells progressively increased with age, while IGF-1 overexpression attenuated the levels of these proteins at all ages. Telomerase activity decreased in aging WT muscle cells but increased in TG, and these changes were secondary to parallel modifications in Akt phosphorylation. A reduction in nuclear phospho-Akt and telomerase activity resulted in telomere shortening and uncapping in WT cardiac cells with aging. Importantly, the senescence and death of CSCs increased with age in WT mice, thus impairing cell homeostasis in the heart. Myocardial IGF-1 over-expression in TG mice prevented CSC senescence and death with age, thus inhibiting ventricular dysfunction. Myocardial IGF-1 overexpression increased nuclear phospho-Akt and telomerase activity, thus preventing cardiac cell senescence and death [27]. Overall, these data suggested that the preservation of functional CSCs and their resulting activation to contribute new cardiac parenchymal cells can prevent/revert the detrimental impact of age on the myocardial tissue.

In a recent study, we isolated human CSCs from biopsies of right atria, obtained from subjects aged 32–86 years with aortic disease, valve disease, coronary artery bypass graft (CABG), or multiple diseases [29]. There were no differences in the number of CSCs isolated from older (>70 years) subjects compared to subjects <70 years. On average, 22 ± 9%, 31 ± 4%, 48 ± 9%, and 56 ± 16% of freshly isolated CSCs expressed p16^INK4A^ isolated from 50–59, 60–69, 70–79, and 80–89 year-old subjects, respectively [29]. Concurrently, we also found an increase in the number of senescence-associated β-galactosidase (senescence-associated-β-galactosidase (SA-β-gal); ~60%) and DNA damage marker γH2AX-positive CSCs (~20%) freshly isolated from old (71–79 years), compared to middle aged (54–63 years) subjects. Moreover, p16^INK4A^-positive CSCs co-expressed γH2AX [29]. The average telomere length of CSCs freshly isolated from old and middle-aged subjects’ hearts were comparable; however, CSCs freshly isolated from old (78–84 years) subjects’ hearts contained a 12% subpopulation with critically short telomeres with a length of <6 kb. CSCs isolated from old (77–86 years) subjects showed a decreased proliferation, clonal amplification, and sphere formation compared to CSCs isolated from middle-aged (34–62 years) subjects [29]. Accordingly, CSCs from old (76–77 years) subjects plated in a cardiomyocyte differentiation medium had a decreased myogenic differentiation potential when compared to middle-aged (47–62 years) subject’s CSCs [29]. Even though CSCs isolated from old hearts showed a decreased proliferation, clonogenicity, and differentiation potential, only ~50% of CSCs are senescent in old myocardiums, so we were able to isolate a functionally cycling-competent CSC population. Indeed, single CSC-derived clones from younger (22–33 years) and old (74–83 years) subjects were indistinguishable in terms of morphology, senescence, multipotency, self-renewing transcript profile, and cardiac differentiation potential. These findings suggest that CSCs become senescent during age in a stochastic, non-autonomous manner. Then, the cycling-competent CSC population (SA-βgal-negative) were compared head-to-head with a senescent (SA-βgal-positive) population for their regenerative potential in an experimental myocardial infarction model in vivo in immunosuppressed NSG mice. The healthy, cycling-competent CSCs showed a significant regenerative and reparative potential on the injured myocardium, resulting in neo-cardiomyogenesis and angiogenesis with improved cardiac function. However, these reparative and regenerative effects were absent when senescent CSCs were injected [29]. Altogether, these findings show that the human heart harbors a CSC compartment that undergoes senescence with age, thus dictating a progressive permanent dysfunction of at least half the cells within this regenerative endogenous cell pool. Therefore, it is still possible to retrieve a healthy, cycling-competent CSC fraction with an increased regenerative and reparative capacity. These data generated the hypothesis that it would be possible to revert CSC aging by selectively eliminating the senescent CSCs and fostering the activation of the healthy aged CSCs. Thus, senescent CSCs affect their microenvironment by decreasing the regenerative potential of the resident stem cell pool. Identifying the molecular mechanisms of cellular senescence in CSCs and selecting a healthy CSC population in aging could be clinically relevant and highly significant to enhance the therapeutic potential of CSC-based repair. In regard to this, nucleostemin (NS), a protein selectively accumulated in the nucleolus of most stem cells, modulates telomere length and is involved in senescence regulation; interestingly, human CSCs show a strong positive correlation between NS expression and telomere length [196]. Accordingly, NS overexpression increases TERT expression, thus preserving telomere length in human CSCs. On the other hand, the loss of function experiments to silence NS expression in CSCs generates a senescent phenotype that is not followed by an increased differentiation of mutated cells [196]. These data, for the first time, suggested that it might be possible to revert human CSC senescence.

## 5. Cardiac Stem/Progenitor Cell SASP

Senescent SA-β-gal-positive CSCs from old human donors showed an increased expression and secretion of SASP factors, including MMP-3, PAI1, IL-6, IL-8, IL-1β, and GM-CSF, compared to non-senescent, SA-β-gal-negative, cycling-competent CSCs [29]. The use of the conditioned medium from senescent CSCs resulted in a decreased proliferation and increased proportion of senescent p16^INK4A^-positive, SA-β-gal-positive, and γH2AX-positive CSCs in relative cell cultures when compared to CSCs treated with conditioned media from cycling-competent CSCs or unconditioned media. These findings, for the first time, showed that the senescent CSCs exhibit an SASP that can negatively impact surrounding cells, causing otherwise healthy and cycling-competent CSCs to lose proliferative capacity and switch to a senescent phenotype [29]. The removal of p16^Ink4a^ senescent cells can delay the acquisition of age-related pathologies in adipose tissue, skeletal muscle, the heart, blood vessels, the lungs, the liver, bone, and eyes [144,145,197,198,199,200,201,202,203,204]. Recent studies have documented the use of senolytic drugs for the selective clearance of senescent cells from “aged” tissues [205]. We therefore tested the potential of four senolytic drugs—Dasatinib (D; an FDA-approved tyrosine kinase inhibitor), Quercetin (Q; a flavonoid present in many fruits and vegetables), Fisetin (F; also a flavonoid), and Navitoclax (N; an inhibitor of several BCL-2 family proteins)—alone and in combination to eliminate and clear senescent CSCs in vitro. Out of all these single senolytic drugs and all their possible combinations, D and Q proved to preserve cycling-competent CSC viability, while senescent CSCs were cleared and induced to selective apoptosis [29]. We next determined whether clearing senescent CSCs using D and Q would abrogate the SASP and its paracrine impact on CSCs. We found that cycling-competent CSCs co-cultured in the presence of senescent CSCs for seven days were decreased in number and proliferation, and they had increased expression of p16^INK4A^, SA-β-gal and γH2AX. The application of D and Q to co-cultures eliminated the senescent CSCs, and seven days later, the cycling-competent CSCs had increased in number, proliferation, and the number of p16INK4A and SA-β-gal CSCs had decreased compared to CSCs that had been in co-culture with senescent CSCs without D and Q. The co-culture of cycling-competent CSCs with senescent CSCs led to an increased secretion of SASP factors into the medium, but the level of SASP factors was reduced with the application of D and Q. Concurrently, the elimination of senescent cells with D and Q treatment in aged (~27 months) mice in vivo decreased cardiomyocyte hypertrophy and fibrosis, activated resident CSCs, and increased the number of small and proliferating cardiomyocytes, as compared to young and old vehicle-treated mice hearts [29]. Overall, these findings documented that senescent human CSCs have an SASP, and the clearance of senescent CSCs using a combination of D and Q senolytics abrogates the SASP and its detrimental senescence-inducing effect on healthy, cycling-competent CSCs (Figure 3). The clearance of senescent cells by senolytic drug treatment in vivo leads to the stimulation of healthy CSCs, resulting in new cardiomyocyte formation that is specific to the aged heart (Figure 3).

The SASP is replete in inflammatory cytokines and chemokines that contribute to ‘inflammaging’ and can directly modulate the immune response [206,207]. CSCs can chemo-attract immune cells and are usually cleared by these immune cells, but SCAPs protect senescent cells from their own pro-apoptotic SASP. During aging and in multiple chronic diseases, senescent cells—and adult senescent stem cells in particular—accumulate in dysfunctional tissues [206,207]. Senescent cells can impede innate and adaptive immune responses [206,207]. Whether an immune system’s loss of capacity to clear senescent cells promotes immune system dysfunction or, conversely, whether immune dysfunction permits senescent cell accumulation are important issues that are not yet fully resolved [207]. Of interest, with aging, Sca-1^pos^/PDGFR-alpha^pos^ cardiac mesenchymal multipotent cells [178,208,209,210] acquires senescence markers and SASP factors involved in immune cell regulation and angiogenesis [132]. The senescence of these multipotent cardiac cells impacts cardiac microenvironment through the modification of their vascular lineage differentiation potentials and the promotion of CCR2-dependent monocyte recruitment [132]. These data open up the tantalizing hypothesis that senescence with aging fosters a skewed differentiation potential within the cardiac multipotent compartment of the heart with the clonal amplification of cells more prone to differentiate towards the endothelial cell lineage and less to the myocyte lineage.

## 6. CSC Senescence and Diabetic Cardiomyopathy

Aging is accompanied by progressive glucose intolerance associated with age-related changes in insulin resistance and pancreatic β-cell function, resulting in a high prevalence of postprandial hyperglycemia, type 2 diabetes mellitus (T2DM), and T2DM-assoaciated macro- and microvascular complications in the elderly population [211,212,213]. Cardiovascular disease (CVD) is the main cause of morbidity and mortality in diabetic patients, accounting for about 80% of all diabetic deaths in North America [214,215]. DM has a dramatic impact on the aging and senescence of different types of adult stem cells [216]. In particular, DM impairs the in vitro proliferative and differentiation potential of adult CSCs, further worsening their senescence phenotype even when compared with CSCs from non-diabetic ischemic patients [217]. DM not only induces a functional decline in resident CSCs, it also reduces cardiac muscle function of diabetic individuals [218]. Indeed, *Ob/ob* and *db/db* mice, common mouse models of obesity and T2DM, show a reduced muscle regeneration after injury by cardiotoxin injection when compared to non-diabetic mice [218].

In a model of insulin-dependent DM, the myocardial accumulation of ROS drives CSC senescence through the expression of p53 and p16^INK4a^ proteins and telomere erosion, which lead to CSC death by apoptosis [219]. The p66^shc^ gene appears to be a significant modulator of these effects because p66^shc^ knockout inhibits CSC senescence and death, preventing the senescent phenotype and the development of cardiac failure by DM [219]. Diabetic p66^shc−/−^ hearts harbor a significantly higher number of resident CSCs when compared to WT diabetic mice, and CSC activation results in an increased cardiomyocyte refreshment with preserved heart function in diabetic p66^shc−/−^ mice. These data have generated the hypothesis that maintaining a healthy and functional the resident pool of CSCs can efficiently offset the detrimental consequences of DM on cardiac tissue [219].

She et al. recently found that diabetes suppresses CSC activation in the heart [220]. In this study, the left coronary artery was permanently ligated to induce a myocardial infarction (MI) in non-diabetic and diabetic rats. Five days later, BrdU incorporation in CSCs showed a significant activation of these cells in the peri-MI zone of non-diabetic rats. However, CSC expansion was significantly reduced in diabetic rats, and the latter was associated with worsened cardiac function at three weeks post-MI. DM was also found to reduce the myocardial expression of SCF expression, together with a reduced phosphorylation of ERK1/2 and p38 MAPK, in the peri-MI of diabetic versus non-diabetic rats [69], thus suggesting that diabetic status diminished SCF expression via a decrease in ERK1/2 and p38 MAPK activation leads to the inhibition of CSC activation [220].

DM determines significant epigenetic alterations that affect stem cell integrity and lead to senescence, in particular through DNA and histone modifications, as well as noncoding RNA (nonprotein coding) regulation by microRNA and long-noncoding RNA [199]. Changes in chromatin conformation were associated by Vecellio et al. with the impaired proliferation, differentiation, and senescent behavior of diabetic CSC [217]. The major identified changes were the hypermethylation of CpG islands, an increased trimethylation of H3K9, H3K27, and H4K20, as well as a decreased monomethylation and acetylation of H3K9 [217]. The latter modifications was found to condense the chromatin and cause a repressive response to hamper the transcription of cell growth genes and genomic stability. Interestingly, the treatment of diabetic CSC with a pro-acetylation compound histone acetylase activator pentadecylidene-malonate 1b (SPV106) reversed chromatin condensation and reverted, at least in part, the senescent phenotype of CSCs by rescuing the proliferation and differentiation potential of diabetic CSCs through an increased acetylation and decreased CpG methylation [217].

T2DM patients at early stages of their disease, while still asymptomatic, show a significant increase in the amounts of circulating and cardiac miR-34a levels when compared to non-diabetic controls [221]. The latter is associated with a specular significant reduction in the expression of the pro-survival protein SIRT1, which is an mRNA specifically targeted for repression by miR-34a. Accordingly, miR-34a is significantly upregulated while SIRT1 is downregulated in adult cardiac muscle cells and CSCs harvested from diabetic hearts; the latter is associated with a higher pro-apoptotic caspase-3/7 activity [221]. However, miR-34 has differential effects depending on the cell context. Indeed, the repression of miR-34a has been found to increase SIRT1 expression in both cardiomyocytes and CSCs; however, the expression of the tumor suppressor p53 protein is further increased in cardiomyocytes with miR-34 inhibition, though it decreased the amount of CSCs. In spite of the increased p53 levels, miR-34a antagonism was found to significantly prevent cardiomyocyte apoptosis by high glucose in culture [221]. On the other hand, miR-34a inhibition significantly reduced proliferation in vitro.

Intriguingly, diabetes has been associated with skewed differentiation potential, which is a typical feature of cell senescence [21]. Indeed, diabetic status was found to alter the fate of CSCs to adipogenesis through the inhibition of the β-catenin/TCF-4 pathway [222].

Altogether, these data postulate the tantalizing hypothesis that the premature cellular senescence and ageing of resident CSCs underpins the development of diabetic heart disease. It will therefore be important to further unravel the major mechanisms underlying CSC senescence where it appears particularly relevant to ascertain how the SASP modulates diabetic CSCs and diabetic hearts [223].

## 7. CSC Senescence and Anthracycline Cardiomyopathy

Cardiovascular and oncologic diseases are the first and second causes of mortality in economically developed countries. These two illnesses can be merged into one group when considering the cardiovascular complications of anticancer therapies [224]. In the last two decades, the importance of the cardiotoxic side effects of anticancer therapies has been increasingly recognized. Accordingly, European and north American experts in the field have put together practice guidelines on the approaches to investigate and manage cancer patients at high risk of cardiovascular complications [224].

Anthracyclines are drugs discovered in Italy more than a half century ago that are still a milestone for many chemotherapy protocols [224]. These drugs are well known to commonly produce cardiotoxic effects [224]. Doxorubicin (DOX), the main member of the anthracycline family, has been widely used for a very long time as a prototypical anticancer drug to induce cardiotoxicity and cardiac dysfunction. The main mechanism of DOX cardiotoxicity has been mainly attributed to the considerable accumulation of ROS and reactive nitrogen species (RNS) in adult cardiac muscle cells [225,226,227]. Beyond the effects on the cardiomyocytes, recent basic science reports have discovered other cardiac cell targets of the DOX-induced cardiotoxicity, identifying the detrimental effects of DOX on the endogenous CSC pool that induce their premature senescence [228,229,230,231,232,233]. It is worth remembering here that DOX is commonly used as an in vitro assay to study cellular and stem cell senescence [29].

In an animal model of anthracycline-induced cardiomyopathy, DOX administration caused ROS activation and DNA damage in resident CSCs with the induction of replicative senescence and apoptotic death [228]. When compared to matched controls with non-cardiovascular diseases, DOX altered the myocardium of treated patients who showed a higher number of CSCs marked by DNA damage and senescence, and in particular by the phosphorylated form of histone H2AX and p16^INK4a^ [29,229]. DOX administration to human CSCs in vitro acutely activates senescent and proapoptotic pathways, advancing the hypothesis that DOX depletes the myocardium of CSC-dependent regenerative potential and making the DOX-injured myocardium more susceptible to chronic damage and failure [229]. Accordingly, DOX-injured human CSCs are unable to foster anatomical and functional regeneration in animals with DOX cardiomyopathy [230,231]. Remarkably, resveratrol, an antioxidant compound and a sirtuin 1 activator, was found to prevent the DOX-induced replicative senescence of CSCs by preventing the excessive accumulation of intracellular ROS and fostering an oxidative stress defense. Moreover, resveratrol treatment fueled the regenerative capacity of CSCs than when intramyocardially injected in damaged hearts, and ameliorated cardiac function and significantly decreased animal mortality [230]. That DOX induces CSC senescence and apoptosis has been independently reproduced. Indeed, DOX increased the number of SA-β-gal-positive CSCs, while human amniotic fluid stem cell secretome pre-treatment was able to significantly decrease DOX-induced CSC damage and death [232]. Additionally, a natural potent antioxidant and polyphenol-rich fraction extracted from citrus bergamot (BPF) was found to significantly prevent DOX-induced cardiotoxicity in rats by reducing DOX-induced ROS upregulation, increasing cardiac cell survival, and restoring protective autophagy while also promoting CSC activation and cardiomyocyte replacement in DOX-injured cardiac tissue [234].

Human epidermal growth factor receptor 2 (HER2)-positive breast cancers are effectively managed by a combination of anthracyclines (e.g., DOX) and trastuzumab (Trz), a humanized anti-HER2/ErbB2 drug [235]. Human CSCs produce and secrete exosomes that exert potent cardioprotective effects when administered in animals with experimental myocardial ischemia [188]. On this basis, animals with DOX/Trz-mediated cardiotoxicity were recently treated with the intravenous administration of CSC-derived exosomes [235]. Animals treated with DOX/Trz administration developed a cardiomyopathy characterized by myocardial fibrosis, CD68^+^ inflammatory cell infiltration, inducible nitric oxide synthase expression, and cardiac dysfunction, all of which were significantly reduced by the injection of CSC-derived exosomes. miR-146a-5p within the CSC-derived exosomes mediated these positive effects [188].

Altogether, these data show that DOX alters CSC function and survival, and it generates the hypothesis that the early toxicity of the resident CSCs caused by the acute exposure to DOX may be responsible for late-onset heart failure in DOX-treated patients. However, whether CSC functional decline by DOX dictates acute and chronic DOX-induced cardiomyopathy remains to be fully ascertained.

## 8. Is CSC Intrinsic Senescence Reversible?

While eliminating senescent cells to rejuvenate tissue and organs through the selection of healthy functional tissue-specific stem cells appears to be valid strategy to prevent/reverse the detrimental effects of senescent-associated aging, it is yet unsubstantiated whether the permanent intrinsic senescence-associated proliferative withdrawal can be reversed [2]. In regard to this, recent findings showing proliferative activity in a senescent cell population implied that senescence is a dynamic, rather than a terminal, phenomenon [236,237,238,239] (Figure 4). The acute inactivation of RB is sufficient to promote proliferation in senescent mouse embryo fibroblasts (MEFs) [237], prevents SAHF accumulation, and cooperates with p53 loss to bypass senescence in human diploid fibroblasts [35,240]. Indeed, the loss of RB inhibits initial replicative senescence in culture; however, these ‘rescued’ senescent proliferative cells eventually have their cell cycle inhibited by an increased phosphorylation of p53, thus resulting in an upregulation of the cell-cycle inhibitor p21 [241]. Therefore, the concurrent knock-down of RB family proteins and p53 is necessary to prevent and revert senescence. Indeed, cells undergoing cell cycle arrest by senescence with low levels of p16^INK4a^ can be converted into proliferative cells through acute p53 inactivation [236]. These data have raised the hypothesis that there is a state of ‘light’ senescence (representing low p16^INK4a^ levels with the incomplete activation of Rb) that is different from a ‘deep/stable’ senescent state, and the factors causing these hypothetical states are still being searched for. The H3K9-active demethylases JMJD2C and LSD1 mediate escape from senescence in fibroblasts or melanocytes [239]. Moreover, fluorescence-based SA-β-gal staining was employed to analyze single cells, and through this methodology the presence of a few senescent cells moving out their cell cycle arrest to duplicate their DNA and divide was discovered [242]. That senescent cells naturally reverted to dividing cells was clearly shown by the evidence that cells labelled by the senescence marker retained it while also showing the incorporation of the marker of DNA replication [242]. Overall, it has been hypothesized that the stability of the senescent state—foremost the permanent cell cycle withdrawal that is typical of senescence status—is mainly driven by the amount and form of senescence-associated chromatin remodeling, and natural senescence reversibility seems to correlate with a less intense epigenetic modification. On the other hand, the thus far accumulated data have shown that natural escape from the senescent state back to a proliferative activity is not a whole conversion back to pre-senescent cell life.

Recent findings have shown that a genetic in vivo strategy for transient reprogramming can ameliorate age-associated senescent hallmarks and extend the lifespan in progeroid mice carrying a Dox-inducible expression cassette for the classical four reprogramming factors, i.e., Oct4, Sox2, Klf4, and c-Myc (OSKM) [244]. Similarly, the transient activation of OSKM in vivo was found to ameliorate recovery from metabolic disease and muscle damage in older wild-type mice [244]. The improvement of senescent features during aging by epigenetic modification through in vivo reprogramming elicited by the short term expression off the OSKM factors lends further support to the main role played by epigenetic regulation of mammalian aging. Recently, the transient expression of OSKMLN as nuclear reprogramming factors, through the release of their relative mRNAs, induced a swift, permanent, and wide improvement and turnaround of cellular aging in human endothelial cells and fibroblasts at the cellular, epigenetic, and transcriptomic levels [245]. Intriguingly, all these epigenetic and transcriptional changes occur before cellular identity is affected by epigenetic reprogramming. Moreover, the transient expression of OSKMLN mRNAs was found to revert the senescence of both mouse- and human-derived skeletal muscle stem cells without abolishing cellular identity and while also restoring their muscle tissue regenerative potential in vivo [245].

The above data show that naturally aged human and mouse cells can be rejuvenated with the restitution of a normal function in both diseased cells and aged stem cells while the cell identity is preserved. It will therefore be interesting to evaluate if epigenetic reprogramming in truly senescent CSCs revert their senescent-associated phenotype and regenerative defects in vitro and in vivo. Of interest, our data (unpublished) show that Bmi-1, a member of the polycomb repressor complex 1 that mediates gene silencing by regulating chromatin structure and that is indispensable for the self-renewal of normal stem cells, is significantly downregulated in aged and senescent CSCs. Restoring Bmi-1 in aged CSCs rescues their replicative defects and self-renewal capacity (unpublished). However, Bmi1 plays a key role as an epigenetic barrier to direct cardiac reprogramming [246]. Bmi1 directly interacts with the regulatory regions of cardiogenic genes, whereby Bmi1 downregulation fosters increased levels of the active tri-methylation at the fourth lysine residue of the histone H3 protein (H3K4me3) and reduced the levels of the repressive ubiquitination of lysine 119 of histone 2A (H2AK119Ub) at cardiogenic loci, resulting in the de-repression of cardiogenic gene expression during cardiomyocyte conversion [246]. Furthermore, Bmi1 deletion replaces the need for Gata4 during cardiomyocyte reprogramming [246]. From here, it can be hypothesized that while Bmi-1 re-expression is necessary to revert CSC senescence, transient non-integrative Bmi-1 overexpression is needed to obtain functional myogenic progeny from senescent CSCs.

## 9. Conclusions

In the 50 years since the role of senescence in aging and disease was first proposed, the cell senescence phenotype has moved from being considered an irreversible and static phenomenon to be recognized as a relevant dynamic and active cellular mechanism in vivo (Figure 4). Adult stem cell senescence contributes to aging and age-related, but also age-unrelated, physiological and pathological processes due to organ dysfunction and disease. In particular, CSCs undergo senescence with age or in response to different stresses, like diabetes and anthracycline therapy, with the activation of the molecular machinery underlying replicative and differentiation defects along with the production and secretion of a detrimental SASP. The clearance of senescent cells by senolytic drugs rejuvenates the myocardial regenerative capacity of aged mice. However, whether CSC senescence is reversible remains to be established. Therefore, it is essential to characterize the mechanisms and functions of senescent CSCs in cardiac aging and disease to design specific, optimal, and nontoxic therapeutic approaches. Concurrently, it is mandatory to obtain a better understanding of the molecular mechanisms of the cell-autonomous and cell non-autonomous features of CSC senescence in vivo to establish which of these two related characteristics can be better targeted. It will also be of particular interest to assess whether and how the senescence and aging of other cardiac cells (cardiac fibroblasts [30] and cardiomyocytes [247], in particular) have a direct effect on CSC senescence during aging and cardiac disease. Nevertheless, all the in vivo biomarkers for senescence, including SA β-gal activity and p16^INK4a^ expression, may be unreliable [243], especially considering the heterogeneity of senescent phenotypes in vivo. Therefore, the single-cell transcriptome profiling of CSCs as a function of age and disease will be necessary to understand the nature of these senescent cells in order to eventually design effective therapeutic approaches.

## Figures and Tables

**Figure 1 cells-09-01558-f001:**
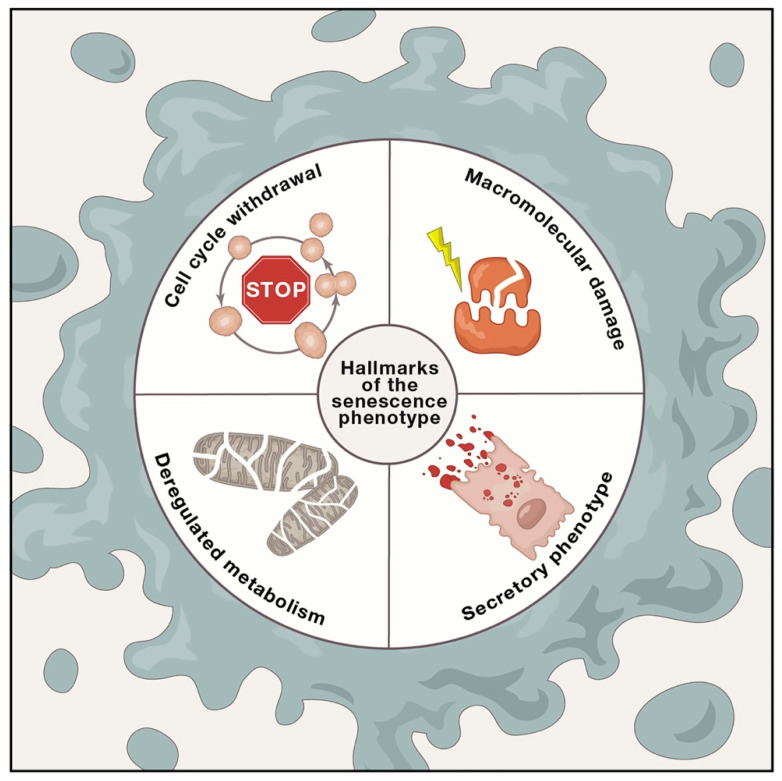
The Hallmarks of the Senescence Phenotype: Senescent cells exhibit the following four interdependent hallmarks: (1) cell-cycle withdrawal, (2) macromolecular damage, (3) the secretory phenotype (SASP), and (4) a deregulated metabolism (see also text). Figure reproduced with permission from Gorgoulis, V. et al. [1].

**Figure 2 cells-09-01558-f002:**
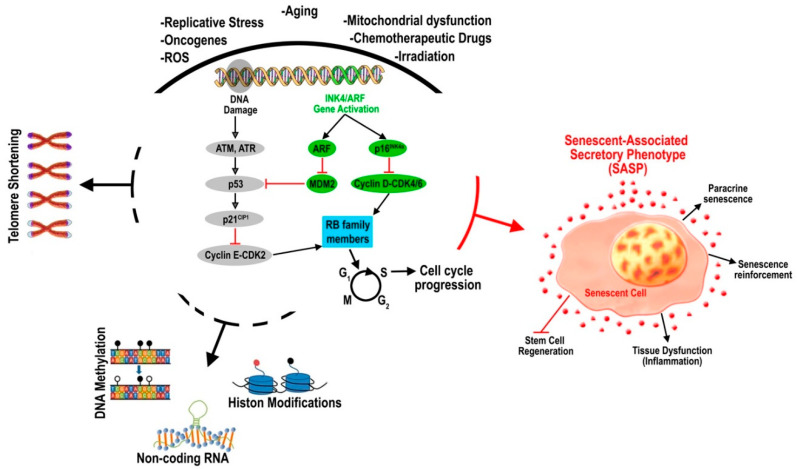
Molecular and Cellular Mechanisms of Senescence: A variety of stressors induce replicative senescence by the induction of the p16^INK4a^/ARF and p53/p21^CIP1^ pathways converging on the Rb family members to block cell growth. Telomere shortening and epigenetic modifications contribute to the senescent status. These molecular modifications progress to the development of the SASP that amplify senescence through cell-to-cell connections that modify tissue homeostasis, repair, and function.

**Figure 3 cells-09-01558-f003:**
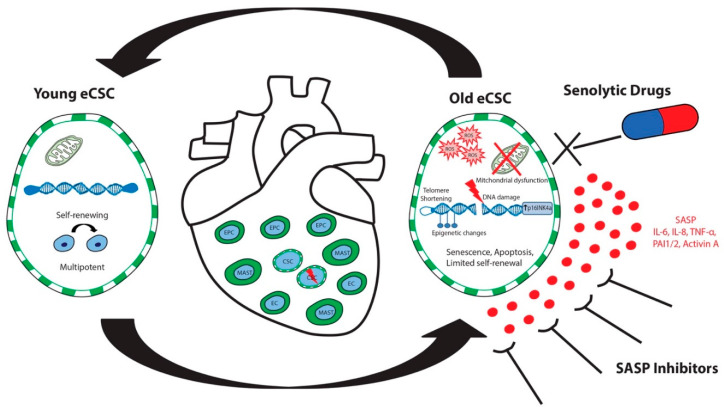
Molecular and Cellular Mechanisms of Senescence: Replicative senescence and SASP characterize the senescent phenotype of aged and dysfunctional CSCs. Senolityc drugs and SASP inhibitors are able to eliminate senescent cells and senescent dysfunctional CSCs, thus favoring the expansion of healthy and functional CSCs.

**Figure 4 cells-09-01558-f004:**
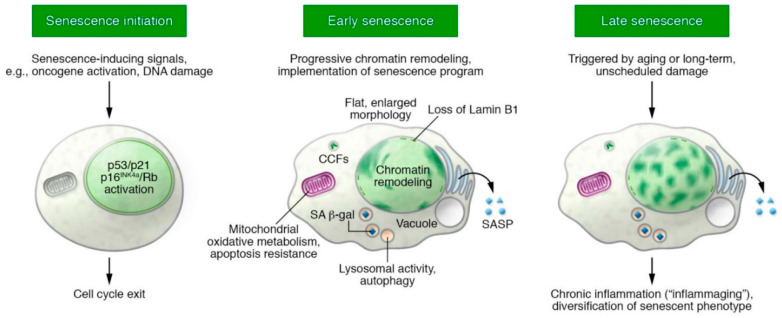
Phenotypic Characteristics of Senescent Cells: Diagram depicting some of the phenotypic alterations associated with senescence initiation, early senescence, and late phases of senescence, which suggest that senescence is a dynamic rather terminal phenomenon. Figure reproduced with permission from Herranz N. and Gil J. [243].

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
