# Peer review of "Targeting Cardiac Stem Cell Senescence to Treat Cardiac Aging and Disease"

_cells, 2020, doi:10.3390/cells9061558_

Round 1
Reviewer 1 Report
The review entitled “Targeting Cardiac Stem Cell Senescence to Treat Cardiovascular Aging and Disease” gives a well-documented overview of the molecular mechanisms driving senescence in the context of cardiac aging or pathological settings.
Minor comments :
1. The authors focused their review on CD31neg CD45neg c-kit+ cardiac stem/progenitor cells which have been described to harbour cardiomyogenic differentiation potential but could also differentiate in non-cardiomyocyte cells such as cells of the vascular lineage (smooth muscle cells or endothelial cells).
Although the role of CSC differentiation in the renewal of cardiomyocyte pool have been intensively debated in the past few years, the regenerative capacity of these cells is well described and could lay on their paracrine activity through direct secretion of growth factors or the production of extracellular vesicles; could the authors comment in the main text (lines 389-391). Authors should also discuss the impact of aging on other cardiac stem/progenitor cells.
Cardiac mesenchymal stromal cells defined by co-expression of PDGFR-alpha and Sca-1 (S+P+) are progenitor cells which could differentiate in cells of the mesodermal lineage (adipocytes, chondrocytes, myofibroblasts) and of the vascular lineage (endothelial cells, vascular smooth cells) (Chong et al. 2011 DOI : https://doi.org/10.1016/j.stem.2011.10.002; Noseda et al. 2015 DOI: https://doi.org/10.1038/ncomms7930, PMID: 25980517 ; Farbehi et al. 2019 DOI: https://doi.org/10.7554/eLife.43882.001; Soliman et al. 2020 DOI : https://doi.org/10.1016/j.stem.2019.12.008). With aging, cardiac S+P+ acquires senescence markers and SASP factors involved in immune cell regulation and angiogenesis. Senescence of cardiac S+P+ impacts cardiac microenvironment by modification of their vascular lineage differentiation potentials and promotion of CCR2 dependent monocyte recruitment (Martini et al. 2019 DOI : https://doi.org/10.1111/acel.13015).
2. In the introduction, the authors highlighted the impact of SASP on surrounding cells and immune network affecting tissue homeostatis during aging (lines 87-91); this point should be developed in the review.
3. Lines 305-306, IL-6 and IL-8 do not act through the CCR2 receptor. Please correct.
4. Line 305, cytokines of the IL-1 family should be cited as cytokine reinforcing senescence as IL-1ß has been demonstrated to induce senescence of human and murine S+P+ progenitor cells (Martini et al. 2019 DOI : https://doi.org/10.1111/acel.13015).
5. Line 308, rodents lack direct homologues of IL-8, Cxcl-3 could have functional similarities with IL-8 in mice but Cxcl15 has low similarities with the human IL-8 gene (CXCL8).
Reviewer 2 Report
- Authors have compiled a vast amount of studies in this review, which makes it unnecessarily long and killing the excitation. The first half of the review is a generalized description of the senescence and aging, in which authors failed to connect with the cardiac senescence or aging. The authors are advised to modify the first half (till page 8) of the review to establish its cardiac connection otherwise remove it.
- Page 1-8: Authors have discussed multiple mechanisms of senescence; however, they have nowhere provided the corresponding cell or tissue type. A few of the examples is like saying “the cell types of the organ in which they operate” (line 94) is not enough; the cell types should be mentioned at every place.
- The review should better start with a brief description of senescence, next describing the phenotypic characteristic of senescence (fig 4), next molecular characterization of cardiac senescence, next SASP, and so on (section 3 onwards).
- The authors are advised to incorporate the adult cardiomyocyte senescence and discuss if there is any effect of SASP on adult CM. It is surprising not to see this topic in the review. Also, the authors should incorporate the studies showing the cardioprotective significance of induced rejuvenation in adult cardiomyocytes.
- Line 47: Authors are suggested to check the sequence of senescence processing. As per the reports, cell cycle arrest should be the last.
- Since cKit is not an independent and universally accepted marker for the CSC population, authors should clearly define their criterion for the selection of the research in this review; specifically, what was the identification criteria for the CSC in the original study. The reader needs to have a clear understanding of the cell type being discussed.
- Lines 468-472, and 502-509: Authors should be more consistent about the use of CSC and CPC. A sudden switching between these two terms creating confusion. Also, it sounds like the author wants to say that these two populations are independent, and CSC only conveys a signal to CPCs to control its senescence status. If it is true, the author should first establish the characteristic of these cell types; otherwise, modify the text to reduce the confusion. For example (lines 468-472): the whole para is about the CSC, and suddenly, at one place, CPC has been used, also in lines 502-509. Fig 3, which has been referred here, there is no mention of CPC.
- The author should clarify whether they want to say cardiac regeneration or cardiac protection as per the best of my knowledge protection better fits with the finding so far.
- The author needs to review their references, as many of them appeared to be a review article, the original studies should be preferred (e.g., line 295: ref. 6,118). Also, there are a few places where provided references are not sufficiently supporting the author’s statement, which should be replaced with the more appropriate ones (e.g., line 300: ref. 118-122).
- Lines 354-357: please provide the reference.
- Lines 157, 167, 620, etc.: There are several typo and grammatical errors in the manuscript, which should be fixed.
- Line 581: The author may want to say cardiovascular and oncologic; please check this.
Round 2
Reviewer 2 Report
The authors have made several changes in the manuscript, which improve the quality of the review. My current concerns are:
- I was not aware of the communication between the authors and the guest editor. However, since the authors also accept the incorporation of the general information about aging and senescence, I would suggest modifying the title accordingly. The present title ‘Targeting Cardiac Stem Cell Senescence to Treat Cardiovascular Aging and Disease’ is misleading as this review is only about the cardiac aging and disease.
- The authors have already discussed the senescence in various organs/cells, including human fibroblasts. However, the authors avoided discussing the senescence mechanism of different cardiac cell types, which may have a direct effect on the CSC. Also, it not convincing to say that ‘article is on cardiac stem/progenitor cell senescence and its role in cardiac aging and age-related and age unrelated diseases.’
Author Response
"Please see the attachment." .
